# DeepPGD: A Deep Learning Model for DNA Methylation Prediction Using Temporal Convolution, BiLSTM, and Attention Mechanism

**DOI:** 10.3390/ijms25158146

**Published:** 2024-07-26

**Authors:** Shoryu Teragawa, Lei Wang, Yi Liu

**Affiliations:** 1School of Software, Dalian University of Technology, Dalian 116024, China; lei.wang@dlut.edu.cn; 2School of Engineering, University of Southern Queensland, 487-535 West Street, Toowoomba, QLD 4350, Australia; yi.liu@unisq.edu.au

**Keywords:** RNA methylation, deep learning, gene expression

## Abstract

As part of the field of DNA methylation identification, this study tackles the challenge of enhancing recognition performance by introducing a specialized deep learning framework called DeepPGD. DNA methylation, a crucial biological modification, plays a vital role in gene expression analyses, cellular differentiation, and the study of disease progression. However, accurately and efficiently identifying DNA methylation sites remains a pivotal concern in the field of bioinformatics. The issue addressed in this paper is the presence of methylation in DNA, which is a binary classification problem. To address this, our research aimed to develop a deep learning algorithm capable of more precisely identifying these sites. The DeepPGD framework combined a dual residual structure involving Temporal convolutional networks (TCNs) and bidirectional long short-term memory (BiLSTM) networks to effectively extract intricate DNA structural and sequence features. Additionally, to meet the practical requirements of DNA methylation identification, extensive experiments were conducted across a variety of biological species. The experimental results highlighted DeepPGD’s exceptional performance across multiple evaluation metrics, including accuracy, Matthews’ correlation coefficient (MCC), and the area under the curve (AUC). In comparison to other algorithms in the same domain, DeepPGD demonstrated superior classification and predictive capabilities across various biological species datasets. This significant advancement in algorithmic prowess not only offers substantial technical support, but also holds potential for research and practical implementation within the DNA methylation identification domain. Moreover, the DeepPGD framework shows potential for application in genomics research, biomedicine, and disease diagnostics, among other fields.

## 1. Introduction

In 1942, the concept of epigenetics was introduced for the first time, defined as the heritable changes in gene expression without altering the DNA nucleotide sequence [1,2]. In the 1950s, MacLintock and Brink experimentally confirmed the validity of epigenetics and were awarded the Nobel Prize in 1983. 

With the advancement of life sciences, various epigenetic mechanisms have gradually been discovered. Numerous epigenetic processes such as acetylation and the methylation of proteins have been identified [3]. A systematic study of epigenetic mechanisms holds crucial significance in deepening the understanding of mysterious biological phenomena and their underlying mechanisms. Since the 1980s, countries worldwide have established layouts and plans for epigenetics research. In 2003, the Human Epigenome Project was launched with the aim of mapping the variable methylation sites of the human genome [4,5]. In 2006, researchers from China, Japan, South Korea, and Singapore convened the first Asian Epigenome Alliance meeting, which has since become a vital platform for communication and collaboration in the development of Asian epigenetic research. Hence, as researchers in the field of bioinformatics, it is imperative to focus on epigenetics research, explore advanced technologies, and strive for productive outcomes in the realm of epigenetic scientific investigation [6].

DNA methylation stands as a significant area of epigenetic study [7]. It has been discovered that DNA methylation commonly occurs at the fifth carbon atom of cytosine in the CpG dinucleotides of DNA sequences. The methyl group (-CH3) from S-adenosyl methionine (SAM) is transferred by DNA methyltransferase (DNMT) to the fifth carbon atom of cytosine, forming 5-methylcytosine [8]. Simultaneously, SAM becomes S-adenosyl homocysteine (SAH) after demethylation. According to incomplete statistical data, there are approximately 28 million CpG sites in the human genome, with the majority of them being methylated, accounting for about 60–80% of the total count [9]. The remaining sites are primarily found in the promoter and exon regions of genes, often defined as CpG islands (CGIs) with GC contents exceeding 50% and lengths ranging from 500 to 1000 bp [10]. Previous research indicates that the impact of DNA methylation frequently results in adverse effects on gene expression [11]. The mechanisms encompass the following aspects: 1. the process of DNA sequence binding with transcription factors is influenced by DNA methylation; 2. methylated CpG sites are likely to bind with CpG-binding proteins, resulting in a heterochromatin structure under the influence of histone-modifying enzymes; 3. The interaction between histone modifiers and DNA methyltransferases promotes the formation of fixed chromatin structures. DNA methylation demonstrates distinct characteristics in different cells, a contrast that becomes pronounced between normal and cancer cells [12]. Studies demonstrate that DNA methylation is implicated in the development of almost all cancers. Furthermore, DNA methylation inhibits the deactivation of transposons, which can be inserted into host DNA and lead to gene inactivation, ensuring the normal function of the body [13,14]. Currently, the three extensively studied types of DNA methylation include N6-Methyladenine (6 mA), 5-Hydroxymethylcytosine (5 hmC), and N4-Methylcytosine (4 mC). Each type has distinct biological functions. N4-Methylcytosine (4 mC) plays diverse roles in regulating DNA replication, distinguishing itself from DNA, maintaining the cell cycle, correcting replication errors, and modulating gene expression. Importantly, 4 mC protects host DNA from degradation by restriction enzymes [15]. 5-Hydroxymethylcytosine (5 hmC), a product of 5-Methylcytosine (5 mC) demethylation, is implicated in neural development and tumorigenesis [16]. N6-Methyladenine (6 mA), a methylated base prevalent in prokaryotes, is primarily involved in host defense mechanisms [17].

In conclusion, as a prevalent epigenetic phenomenon, DNA methylation plays a vital role in maintaining the stability of genetic information concerning gene expression regulation, chromatin structural variations, and transposon inactivation. Consequently, the establishment of efficient DNA methylation prediction methods not only advances biological research, but also holds significant implications for disease diagnosis, classification, and personalized treatment development [18,19].

Although detecting the DNA methylation status using experimental means can yield more accurate predictive results, the high economic cost limits the advancement of DNA methylation research to some extent [20]. The widespread use of computational models supplements DNA methylation research. Predicting DNA methylation using machine learning algorithms has become a focal point in the field of bioinformatics, offering a convenient approach to exploring whole-genome DNA methylation patterns across multicellular and multitype tissues. Although many traditional machine learning methods have been employed for DNA methylation prediction [21,22], these methods struggle to effectively extract feature information, necessitating enhancements in the predictive accuracy. For example, 4 mCPred [15] uses an SVM to predict 4 mC DNA methylation, and MM-6 mAPred [17] employs the Markov algorithm to predict 6 mA DNA methylation. Compared with these traditional algorithms, the algorithm presented in this paper could extract greater feature information such as multidimensional structural information. Additionally, DeepPGD could autonomously learn feature representations from the input, thereby reducing the reliance on manual feature engineering.Deep learning models can automatically extract complex features from data without the need for manual feature engineering. Additionally, these models can efficiently and accurately process and analyze large volumes of data [18,19,20].

In this paper, the model incorporated a CNN block with an attention mechanism and a BiLSTM block with an attention mechanism. The CNN block used convolutional kernels of different sizes to increase the dimensionality of the structural feature extraction. The BiLSTM block extracted sequential information, enhancing the diversity of the feature extraction by the network. The DeepPGD model could effectively classify methylated and nonmethylated DNA. This motivation propelled us to further investigate this topic using deep learning technology. Residual network deep learning models exhibit robust feature learning capabilities, effectively extracting highly complex and crucial nonlinear features, thereby improving the model’s predictive performance for the DNA methylation status. To enhance the accuracy of DNA methylation prediction, we applied our deep learning model to a large methylation dataset and compared it with existing deep learning prediction methods. The results indicated that our approach enhanced the accuracy of DNA methylation prediction, thus exhibiting significant implications for the advancement of deep learning technology research in DNA methylation prediction.

Predictive algorithms for DNA methylation may bring about a series of biological implications. These include:

Understanding epigenetic regulatory mechanisms: DNA methylation is a crucial epigenetic modification that plays a key role in gene expression and cell differentiation. Predictive algorithms can help reveal patterns of DNA methylation across different cell types and physiological states, thereby enhancing our understanding of epigenetic regulatory mechanisms.

Discovery of biomarkers: DNA methylation plays an important role in the onset and progression of diseases such as cancer and neurodegenerative disorders. Predictive algorithms can assist in the identification of DNA methylation patterns associated with specific diseases, aiding the discovery of new biomarkers and improving the early diagnosis and treatment of diseases.

Guidance for genome editing and therapy: DNA methylation plays a significant role in genome stability and functional gene regulation. Predictive algorithms can help identify and interpret the location and function of DNA methylation in the genome, providing guidance and a reference for genome editing and therapy.

## 2. Results

In order to investigate the practical performance of the model proposed in this study, publicly available datasets comprising 10 distinct DNA methylation datasets were employed as experimental materials. The experimental results were comprehensively compared with those of benchmark algorithms, revealing substantial advantages. This chapter provides a visual demonstration of the comprehensive outstanding performance of the proposed algorithm, emphasizing its commendable predictive capabilities across multiple evaluation metrics.

### 2.1. Experimental Results

Through experimentation, DeepPGD exhibited several advantages across diverse datasets, particularly in terms of Matthews’ correlation coefficient (MCC) and the area under the curve (AUC). The advantages of DeepPGD are highlighted below (Table 1, Table 2 and Table 3).

Higher Matthews’ correlation coefficient (MCC): DeepPGD demonstrated relatively higher MCC values across most biological species. Through experiments, it was found that the DeepPGD algorithm outperformed the comparative algorithms in 9 out of 10 datasets, except for the 5 hmC *H. sapiens* dataset. Particularly significant advantages were observed in the experimental results for the 4 mc *C. equisetifolia* and *S. cerevisiae* datasets. The iDNA-ABT [23] algorithm performed best on the 5 hmC *H. sapiens* dataset. However, the iDNA-MS [24] algorithm exhibited a relatively mediocre performance across all datasets. The MCC metric evaluates the predictive performance of a classification model, being particularly useful for imbalanced datasets. Elevated MCC values signified that DeepPGD maintained a well-balanced performance in classifying positive and negative instances, potentially conferring advantages in reliability and robustness for DNA methylation prediction tasks.

Higher area under the curve (AUC): DeepPGD presented higher AUC values across most biological species. The AUC measures the overall performance of a classification model across different thresholds and is often employed to assess the classifier’s discriminative ability. Increased AUC values suggested that DeepPGD possessed a strong discrimination ability between positive and negative instances, potentially enhancing its capacity to identify DNA methylation sites.

Comprehensive performance superiority: DeepPGD demonstrated superiority across various metrics using multiple biological species, indicating a favorable performance across multiple aspects. Its comprehensive performance superiority implied that DeepPGD could offer stable and proficient performance across diverse data contexts.

Adaptability to biological species: DeepPGD excelled across different biological species, implying a certain degree of adaptability to various organisms. The ability to adapt to different biological species is crucial when analyzing diverse types of DNA methylation datasets, and DeepPGD’s performance in this aspect enhanced its appeal.

Overall, based on these datasets, the DeepPGD algorithm consistently exhibited superior performance, particularly in terms of Matthews’ correlation coefficient and the area under the curve. The iDNA-ABT algorithm also demonstrated competitive performance across most biological species, while iDNA-MS appeared to exhibit a comparatively lower performance in most performance metrics. 

### 2.2. Box-Plot Analysis

Based on the box plots derived from the experimental data, the following analytical conclusions were drawn. In terms of accuracy (ACC), the box plots demonstrated that the DeepPGD algorithm exhibited higher accuracy across various biological species datasets, markedly surpassing the iDNA-ABT and iDNA-MS algorithms. Although iDNA-ABT and iDNA-MS displayed certain fluctuations in accuracy, their overall levels remained comparatively lower. Concerning Matthews’ correlation coefficient (MCC), the DeepPGD algorithm significantly outperformed the other two algorithms in terms of MCC values for most datasets, indicating its superior predictive performance in classification tasks. Meanwhile, the distribution ranges of the MCC values for iDNA-ABT and iDNA-MS were wider, suggesting potential instability. In the context of the area under the curve (AUC), the DeepPGD algorithm demonstrated elevated AUC values for most datasets, indicating larger areas under the ROC curves and stronger classification capabilities. In contrast, the AUC values for iDNA-ABT and iDNA-MS were generally relatively lower.

In consideration of these metrics, the following conclusions could be drawn. The DeepPGD algorithm possessed a distinct advantage in DNA methylation recognition tasks, excelling in accurate classification and prediction. It consistently exhibited outstanding performance across the accuracy, MCC, and AUC evaluation metrics (Figure 1, Figure 2 and Figure 3).

### 2.3. Ablation Experiments

To gain a deeper understanding of the algorithm’s performance, we conducted ablation experiments on one representative dataset from each of the 4 mC, 5 hmC, and 6 mA categories. Based on the ablation experiment results, several analyses were derived (Figure 4).

Impact on accuracy (ACC): The ablation experiment results revealed that the DeepPGD model consistently achieved higher accuracy for all three biological species datasets when both BiLSTM and a TCN were used. This underscored the crucial role of BiLSTM and the TCN in enhancing accuracy within the DeepPGD model. Conversely, removing either the BiLSTM or TCN component from the model, as indicated by the experimental results, led to decreased accuracy for these datasets (Figure 5).

Impact on Matthews’ correlation coefficient (MCC): Similarly, the experimental results demonstrated that DeepPGD achieved the highest MCC values for all three biological species datasets. The ablation experiments highlighted that the removal of BiLSTM or the TCN led to reduced MCC values. This indicated the pivotal role of BiLSTM and the TCN in enhancing the robustness and predictive performance of the classification model (Figure 6).

Impact on area under the curve (AUC): The AUC is a crucial metric for assessing the predictive performance of classification models. The experimental results indicated that DeepPGD attained the highest AUC values for two of the three biological species datasets when both BiLSTM and TCN were used. However, for one dataset, omitting the TCN resulted in higher AUC values, which consequently improved the performance of the classification model for that specific dataset (Figure 7).

Overall, the ablation experiment results underscored the significant roles of BiLSTM and a TCN within the DeepPGD model in improving key metrics such as accuracy, MCC, and the AUC. This further validated the superior performance of the DeepPGD algorithm in DNA methylation recognition tasks and highlighted the interdependence and importance of different components within deep learning models. Consequently, the integration of BiLSTM and a TCN into the deep learning framework enhanced the accurate identification capability for DNA methylation.

### 2.4. Cross-Species Prediction of DNA Methylation Correlation

In order to understand the correlation between DNA methylation among different species, this study conducted cross-testing using DNA methylation datasets from various species. The experiments revealed that DNA methylation among different species may be correlated and could be predicted using DNA methylation data from other species, although the effectiveness was limited. Among all test results, training and testing using data from the same species yielded the best predictive performance. However, in cross-prediction scenarios, the results were not always positive. For instance, the 6 mA_*A. thaliana* training set could predict the 6 mA_*D. melanogaster* test set relatively well, but the 6 mA_*T. thermophile* training set performed poorly when predicting the 6 mA_Xoc BLS256 test set. Considering these experimental results, predicting the DNA methylation of unknown species using existing species’ DNA methylation data may require complex decision-making (Figure 8).

## 3. Discussion

Deep learning has rapidly advanced and made significant strides across various fields, such as natural language processing and computer vision [25,26,27]. The exploration of machine learning and deep learning methods for DNA methylation classification has been a prominent avenue of research, demonstrating their effectiveness over standard biological and statistical approaches. A predominant focus of these studies has been on binary classification, determining whether specific sequences belong to DNA methylation sites [23,24,28,29]. 

Deep-4mCGP [30], a model approached from a 4 mC classification perspective, employs gradient-boosting decision tree for the feature selection. Subsequently, it feeds the combination of sequences and features into a convolutional neural network (CNN) for 4 mC classification. Mouse4mC-BGRU [31] employs adaptive embedding to extract features from sequences and utilizes bidirectional gated recurrent units (BiLSTMs) for the encoding, adhering to a conventional neural 4 mC classification network. DNC4mC-Deep [32] attempts embedding using a cross-species dataset and applies various encoding methods, including the dinucleotide composition (DNC) and the trinucleotide combination (TNC) position. Finally, it uses a modified neural network for 4 mC classification. Prior to applying a CNN model, Deep-4 mCW2V encodes DNA sequences using word embeddings to learn and accurately identify 4 mC sites. Concerning 5 hmC, iRNA5hmC [33] employs position nucleotide binary carriers as features, utilizes a two-stage optimization method for the feature selection, and then employs support vector machines (SVMs) for the classification. Regarding typical RNA methylation, a well-described study [34] summarizes several classification methods, including binary encoding (e.g., specific-position k-mers) and combination encoding (e.g., k-spacing nucleotide pair frequency (KSNPF)). Researchers have employed SVMs, random forests (RFs), CNNs, and other machine learning models for classification. In 6 mA prediction research, Le and Ho [28] designed a complex predictor structure comprising pretrained transformers and CNNs. i6mA-Vote [35] employs one-hot encoding and five selected basic machine learning models for voting. MGF6mARice [36] transforms original sequences into a simplified SMILES format and employs graph convolutional networks (GCNs) for the encoding and classification.

However, whether traditional machine learning methods or deep learning methods, most still heavily rely on manual approaches to train classifiers for model inputs, necessitating researchers to possess prior knowledge. Furthermore, these methods struggle to be universally applicable across all species. Additionally, the aforementioned methods are only designed for a specific methylation type, or some methods are only tailored to a particular species. Therefore, there is an urgent need for a universal method to identify cross-species DNA methylation sites. iDNA-MS [24] initially configured samples through three sequences to encode features and then employed a RF to identify DNA methylation sites for 5 hmC, 6 mA, and 4 mC. However, the experimental results showed that its algorithm performance was suboptimal, leaving room for improvement.

Given the aforementioned backdrop, this paper introduces an innovative deep learning framework named DeepPGD, aiming to address the issue of DNA methylation identification. To overcome the complexity of DNA sequences and the importance of sequence features, this framework adopted a dual residual structure that combined convolutional networks (CNNs) and bidirectional long short-term memory (BiLSTM) networks, thus wielding formidable feature extraction capabilities. The introduction of a dual residual structure further enhanced the model’s depth, aiding the model to learn more abstract and higher-level features from raw DNA sequences. This structure gradually constructed multilayered feature representations, allowing the model to better comprehend the hierarchical structure of the data. A CNN itself excels at capturing features of different scales, while a BiLSTM can capture long-term dependencies within sequences. By combining them, DeepPGD proved capable of multiscale feature learning, thereby providing a more comprehensive understanding of DNA sequence characteristics.

## 4. Materials and Methods

### 4.1. Dataset

The datasets utilized in this study were the same as those in iDNA-MS [24]. This study employed a total of 10 datasets, representing the following three types of DNA methylation site: 4 mC, 5 hmC, and 6 mA. The 4 mC category consisted of the following three datasets: 4 mC_*C. equisetifolia*, 4 mC_*F. vesca*, and *S. cerevisiae*. The 5 hmC category included the dataset 5 hmC_*H. sapiens*. The 6 mA category comprised the following six datasets: 6 mA_*A. thaliana*, 6 mA_*C. elegans*, 6 mA_*D. melanogaster*, 6 mA_*H. sapiens*, 6 mA_Tolypocladium, 6 mA_*T. thermophile*, and 6 mA_Xoc, BLS256. All datasets shared a common DNA sequence length of 41 nt.

### 4.2. Evaluation Criteria

In this research, the investigators utilized the AUC (area under the curve), ACC (accuracy), and MCC (Matthews’ correlation coefficient) as the primary evaluation criteria to assess the performance of predictive models based on DNA methylation data. DNA methylation prediction is a significant binary classification problem that involves predicting the presence or absence of methylation sites within DNA molecules. The following elaboration and explanation provide further insights into these evaluation criteria.

AUC (area under the curve): In DNA methylation prediction, the AUC is used to measure the balance between the true-positive rate and false-positive rate of a model at various classification thresholds. A high AUC value indicates that the model can accurately distinguish between positive and negative samples, thereby demonstrating a good predictive performance.

ACC (accuracy): ACC in DNA methylation prediction measures the overall classification accuracy of a model. The accurate classification of positive and negative samples is crucial for the task of DNA methylation prediction as the predictive outcomes can have significant implications in biological research and medical diagnosis.
(1)ACC=(TP+TN)/(TP+TN+FP+FN)

MCC (Matthews’ correlation coefficient): MCC is a comprehensive metric that takes into account true positives, true negatives, false positives, and false negatives. In DNA methylation prediction, MCC offers a more holistic performance evaluation. In particular, when dealing with imbalanced datasets, MCC better reflects the model’s performance, aiding researchers to accurately assess the model’s performance across different classification scenarios.
(2)$$MCC=\frac{TP\times TN-FP\times FN}{\sqrt{\left(TP+FP)(TP+FN)(TN+FP)(TN+FN\right)}}$$

The combined application of these evaluation criteria helps researchers to gain a comprehensive understanding of the performance of DNA methylation prediction models, enabling more accurate model selections and optimization decisions. In the field of bioinformatics, these evaluation metrics hold significant significance in verifying model reliability, applicability, and their role in uncovering gene regulation and disease mechanisms.

### 4.3. RNA Representation Method

This study encompassed a total of ten datasets, each containing DNA sequences of a fixed length. During the data preprocessing phase, the 3-mer technique was employed to process the original DNA sequences, resulting in a reduction in the sequence length. Subsequently, to achieve a uniform sequence length, a zero-padding strategy was applied at the end of the DNA sequences, extending the length to 48 base pairs.

In the subsequent stages of processing, the word embedding technique was employed to convert DNA sequences into corresponding matrix representations. In this process, each DNA motif was mapped to a sixteen-dimensional embedding space, facilitating the conversion from sequence data to continuous vector representations. The resultant matrix possessed dimensions of 1 × 48 × 16, providing a rich informational foundation for the subsequent in-depth analysis and model construction (Figure 9).

### 4.4. Residual MLP Block in DeepPGD

In the DeepPGD model, the multilayer perceptron (MLP) layer could be viewed as an operation for feature compression and mapping. The role of the MLP layer lay in abstraction and encoding. Through the introduction of nonlinear activation functions, the MLP could map original features to a higher-level abstract representation space, aiding the model to comprehend and distinguish patterns and relationships within the data. Furthermore, in this context, an attention mechanism was introduced, enabling the model to learn feature disparities between the MLP layer and the original DNA sequences (Figure 10).

### 4.5. TCN Block in DeepPGD

A temporal convolutional network (TCN) is a neural network architecture designed specifically for sequence data, adept at capturing relationships and patterns within DNA sequences. This is crucial for methylation detection tasks, as the methylation state in DNA sequences is often influenced by adjacent base pairs. A TCN utilizes causal convolutions to ensure that predictions at any given point in the sequence depend only on current and past information, preserving the sequence order and avoiding information leakage from future points.

In this study, a TCN model employing causal convolutions was used to extract sequence features from DNA sequences. Note that while TCN typically involves various components such as dilated convolutions, this study specifically utilizes causal convolutions. Causal convolutions ensure that each convolution operation only depends on current and past information, which is crucial for maintaining the sequence order of the data. The model incorporated multiple causal convolutional kernels of varying sizes within each block to capture structural information across different spans of the sequence. This design enables the TCN to effectively handle long-range dependencies by preserving the sequence while considering distant base pairs (Figure 11 and Figure 12).

MHA (multihead attention) is an important mechanism used to calculate the weighted sum of the key vector according to the query vector where the weight is determined by the similarity between the query vector and the key vector. The formulas are as follows:(3)$$Q_{i}=QW_{i}^{Q},K_{i}=KW_{i}^{K},V_{i}=VW_{i}^{V},i=1,...,h$$
(4)$$head_{i}=Attention(Q_{i},K_{i},V_{i}),i=1,...,h$$
(5)$$MultiHead(Q,K,V)=Concat(head_{1},...,head_{h})W^{O}$$
where Q, K, V represent the query matrix, key matrix, and value matrix, respectively; $W_{i}^{Q},W_{i}^{K},\;and\;W_{i}^{V}$ represent the weight matrices of the query matrix, key matrix, and value matrix, respectively; $W^{O}$ represents the output weight matrix; h represents the number of heads; $head_{i}$ represents the output of the i-th head; and Concat represents the concatenation operation. 

### 4.6. BiLSTM Block in DeepPGD

Bidirectional long short-term memory (BiLSTM) represents a variant of a recurrent neural network (RNN) that exhibits bidirectional properties, enabling the simultaneous consideration of forward and backward contextual information. In the context of methylation detection tasks, each DNA base pair is regarded as a point in the sequence, allowing BiLSTM to effectively capture sequence dependencies. The BiLSTM model performs computations in both the forward and backward directions. In the forward computation, the model starts with the first base pair in the sequence and sequentially calculates the hidden state for each subsequent base pair. In the backward computation, the model starts from the last base pair and calculates the hidden states in reverse order. The hidden states from both directions are combined through concatenation or addition, yielding comprehensive contextual information. These hidden states can be interpreted as feature representations of each base pair in the sequence, capturing patterns and relationships. These feature representations serve as the foundation for the methylation state prediction in subsequent layers (Figure 13).

In this study, a BiLSTM residual model augmented with a multihead cross-attention mechanism was employed to extract sequence features. Specifically, three BiLSTM units were employed to extract sequence features from DNA sequences, enabling the extraction of complex hierarchical features. The formulas for the LSTM calculation are as follows:(6)$$f_{t}=\sigma (W_{f}\cdot [h_{t-1},x_{t}]+b_{f})$$
(7)$$i_{t}=\sigma (W_{i}\cdot [h_{t-1},x_{t}]+b_{i})$$
(8)$${\tilde{C}}_{t}=tanh(W_{C}\cdot [h_{t-1},x_{t}]+b_{C})$$
(9)$$C_{t}=f_{t}*C_{t-1}+i_{t}*{\tilde{C}}_{t}$$
(10)$$o_{t}=\sigma (W_{o}\cdot [h_{t-1},x_{t}]+b_{o})$$
(11)$$h_{t}=o_{t}*tanh(C_{t})$$

## 5. Conclusions

In this paper, we introduced DeepPGD, an innovative deep learning model designed to address the challenge of DNA methylation recognition. By integrating the attention mechanism of a transformer with MLP layers and by incorporating the feature extraction of a TCN and BiLSTM, DeepPGD demonstrated remarkable performance using DNA methylation datasets from multiple biological species. Through its adept ability to efficiently capture both structural and sequence features of DNA sequences, DeepPGD surpassed conventional methods in metrics such as accuracy, MCC, and AUC. Based on the experimental findings of this study, diversity in feature extraction using models may contribute to an improvement in model performance. Therefore, we believe that further exploration into the diversity of effective feature extraction using models could be a future direction in this field of research. This accomplishment provides a robust tool to advance DNA methylation research. Despite certain challenges that remain, this study provides a new direction for the application of deep learning within the field of bioinformatics.

## Figures and Tables

**Figure 1 ijms-25-08146-f001:**
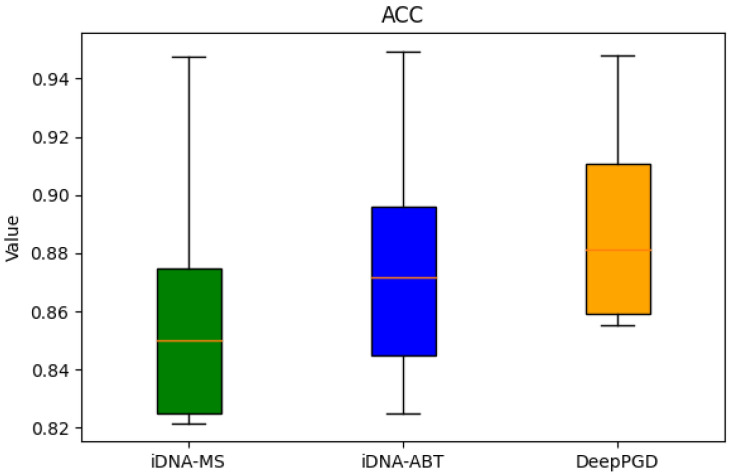
Box-plot analysis based on accuracy.

**Figure 2 ijms-25-08146-f002:**
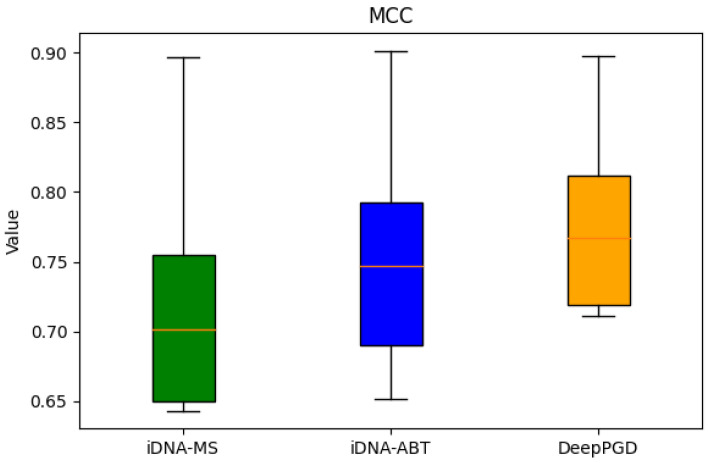
Box-plot analysis based on MCC.

**Figure 3 ijms-25-08146-f003:**
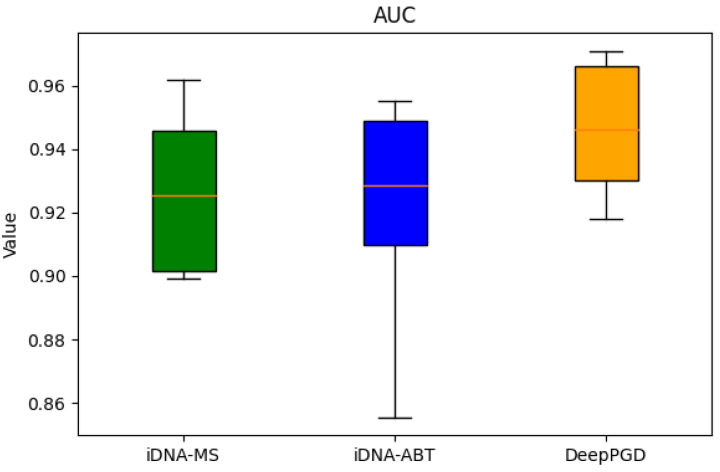
Box-plot analysis based on the AUC.

**Figure 4 ijms-25-08146-f004:**
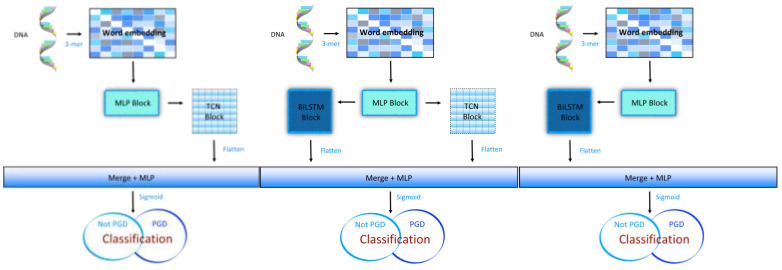
The introduction of models in the ablation experiments. The left part of the figure shows the original model with the LSTM module removed. The middle part of the figure shows the original model. The right part of the figure shows the original model with the TCN module removed.

**Figure 5 ijms-25-08146-f005:**
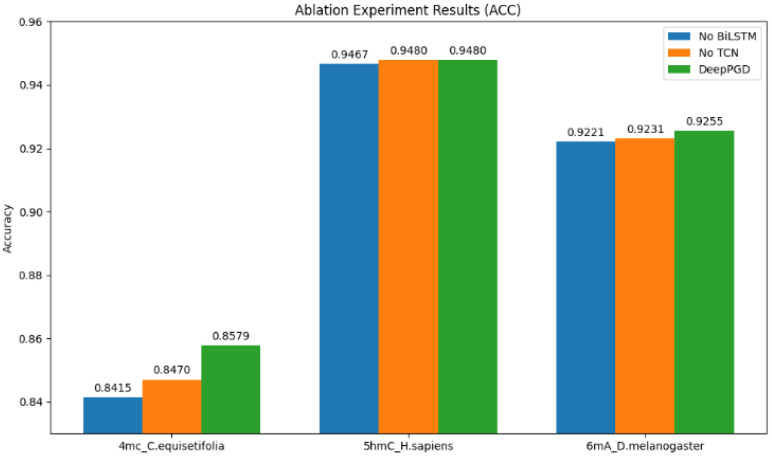
Comparison of accuracy performance in ablation experiments.

**Figure 6 ijms-25-08146-f006:**
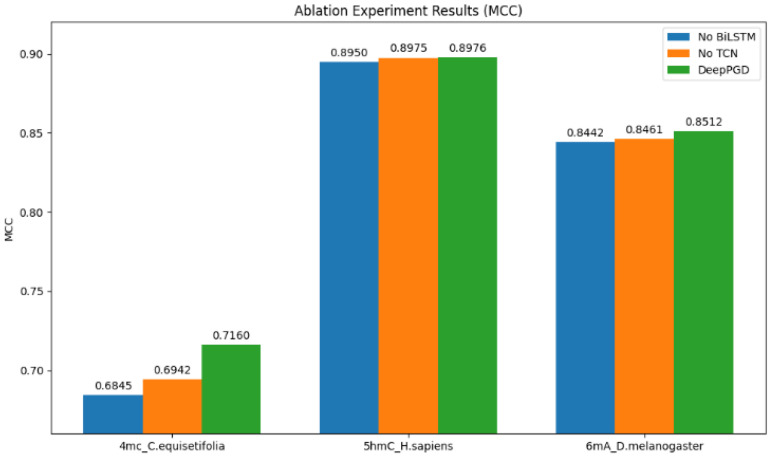
Comparison of MCC performance in ablation experiments.

**Figure 7 ijms-25-08146-f007:**
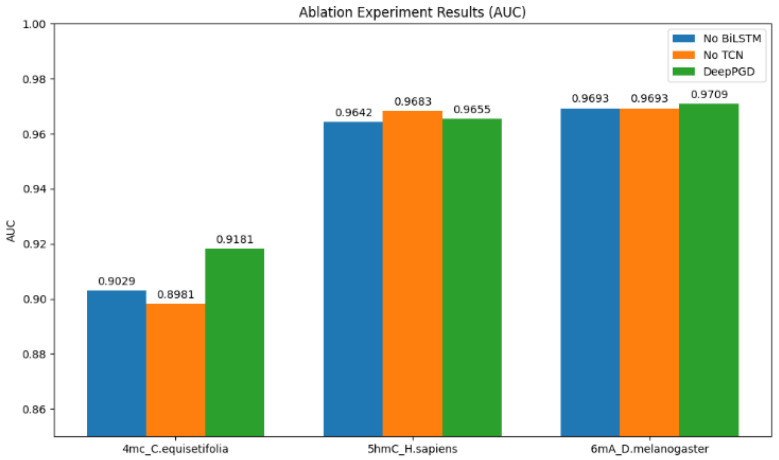
Comparison of AUC performance in ablation experiments.

**Figure 8 ijms-25-08146-f008:**
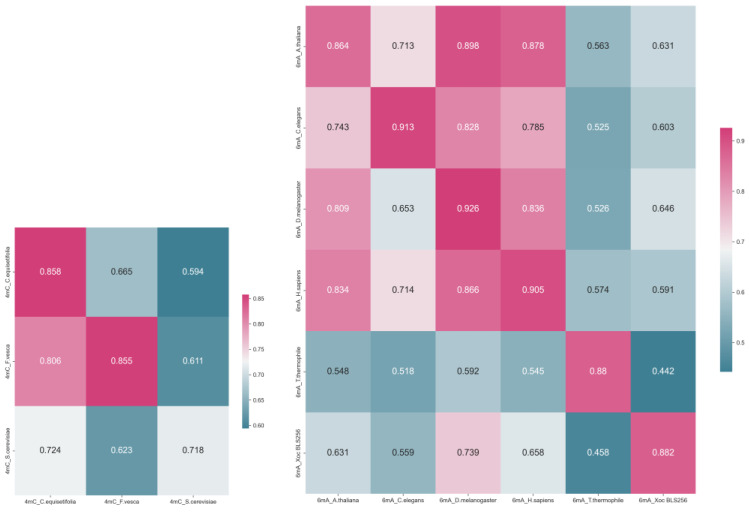
The results of cross-prediction of DNA methylation among different species, with the vertical axis representing the training set and the horizontal axis representing the test set (ACC).

**Figure 9 ijms-25-08146-f009:**
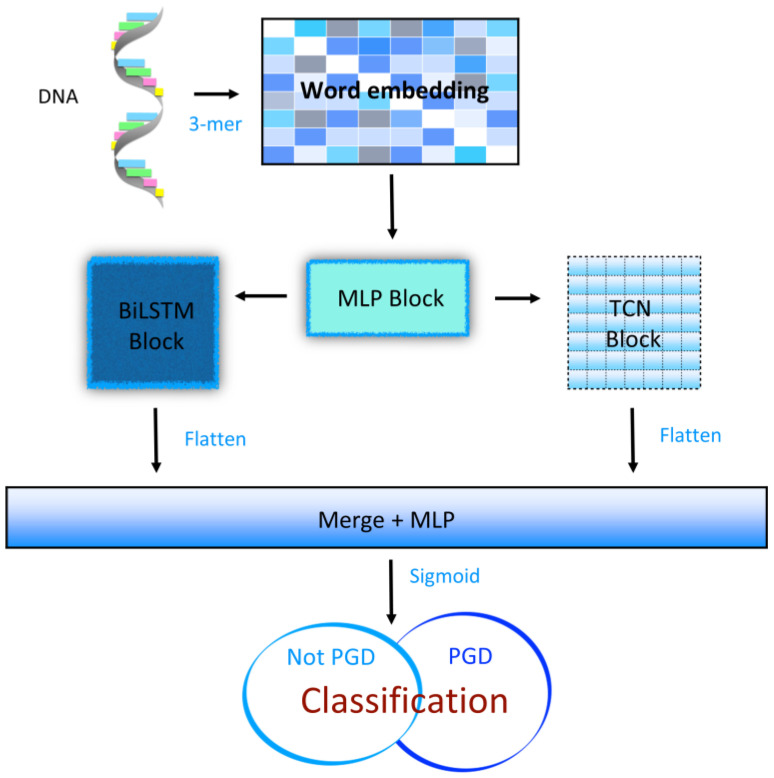
The structure of DeepPGD.

**Figure 10 ijms-25-08146-f010:**
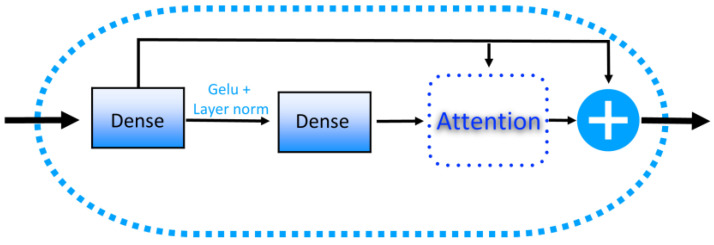
The structure of the MLP block.

**Figure 11 ijms-25-08146-f011:**
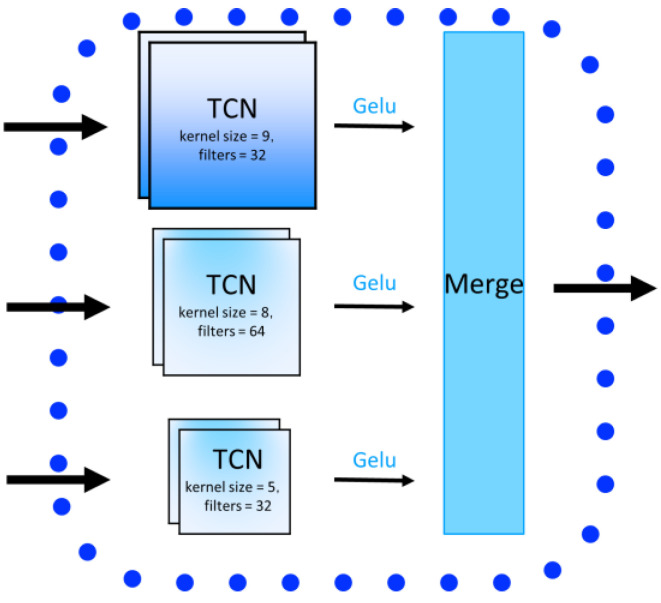
The basic structure of the CNN block.

**Figure 12 ijms-25-08146-f012:**
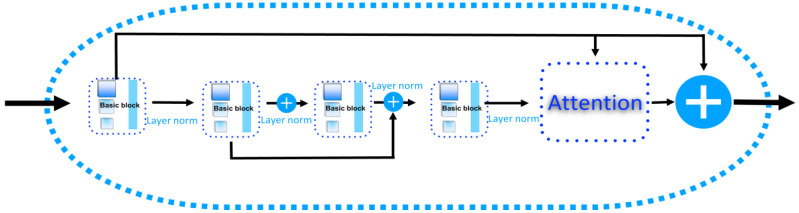
The structure of the CNN block.

**Figure 13 ijms-25-08146-f013:**
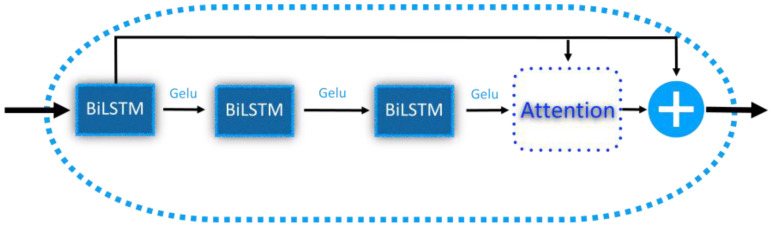
The structure of the BiLSTM block.

**Table 1 ijms-25-08146-t001:** ACC comparison of each method.

	4 mc	4 mc	4 mc	5 hmC	6 mA	6 mA	6 mA	6 mA	6 mA	6 mA
	*C. equisetifolia*	*F. vesca*	*S. cerevisiae*	*H. sapiens*	*A. thaliana*	*C. elegans*	*D. melanogaster*	*H. sapiens*	*T. thermophile*	Xoc BLS256
**i** **DNA-MS**	0.7262	0.8217	0.6962	0.9475	0.834	0.8549	0.9027	0.8799	0.8602	0.8449
**iDNA-ABT**	0.8251	0.842	0.7027	**0.9492**	0.8538	0.8903	0.9122	0.898	0.874	0.8694
**DeepPGD**	**0.8579**	**0.8554**	**0.7179**	0.948	**0.8636**	**0.9127**	**0.9255**	**0.9045**	**0.8802**	**0.8824**

The best value in each column is in bold.

**Table 2 ijms-25-08146-t002:** MCC comparison of each method.

	4 mc	4 mc	4 mc	5 hmC	6 mA	6 mA	6 mA	6 mA	6 mA	6 mA
	*C. equisetifolia*	*F. vesca*	*S. cerevisiae*	*H. sapiens*	*A. thaliana*	*C. elegans*	*D. melanogaster*	*H. sapiens*	*T. thermophile*	Xoc BLS256
**i** **DNA-MS**	0.452	0.6433	0.395	0.8966	0.6697	0.7099	0.805	0.7623	0.7342	0.693
**iDNA-ABT**	0.6517	0.6842	0.4064	**0.9009**	0.7088	0.7808	0.8244	0.796	0.754	0.7394
**DeepPGD**	**0.716**	**0.711**	**0.436**	0.8976	**0.7273**	**0.8121**	**0.8512**	**0.8091**	**0.7688**	**0.7649**

The best value in each column is in bold.

**Table 3 ijms-25-08146-t003:** AUC comparison of each method.

	4 mc	4 mc	4 mc	5 hmC	6 mA	6 mA	6 mA	6 mA	6 mA	6 mA
	*C. equisetifolia*	*F. vesca*	*S. cerevisiae*	*H. sapiens*	*A. thaliana*	*C. elegans*	*D. melanogaster*	*H. sapiens*	*T. thermophile*	Xoc BLS256
**i** **DNA-MS**	0.79	0.8991	0.7612	0.962	0.9093	0.9311	0.962	0.9507	0.926	0.9251
**iDNA-ABT**	0.8555	0.907	0.7537	0.9553	0.9184	0.9433	0.9544	0.951	0.931	0.9261
**DeepPGD**	**0.9181**	**0.9285**	**0.7763**	**0.9655**	**0.9354**	**0.9662**	**0.9709**	**0.9664**	**0.943**	**0.9497**

The best value in each column is in bold.

## Data Availability

Data availability statements are available at https://github.com/FROZEN160/DeepPGD (accessed on 24 June 2024).

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
