# Peer review of "DeepPGD: A Deep Learning Model for DNA Methylation Prediction Using Temporal Convolution, BiLSTM, and Attention Mechanism"

_ijms, 2024, doi:10.3390/ijms25158146_

Round 1
Reviewer 1 Report
Comments and Suggestions for Authors
In the manuscript “DeepPGD: A Deep Learning Model for DNA Methylation Prediction Using Temporal Convolution, BiLSTM, and Attention Mechanism”, Teragawa and colleagues employ a deep learning strategy to predict DNA methylation sites in a variety of genomes. Characterising such sites is important to understand gene regulation, particularly in the absence of experimental data. Some comments:
1. The authors introduce the topic and the literature review relatively well but there is no explanation/definition about the DNA methylation types (4mc 5hmC 6mA).
2. I would bring the materials and methods prior to results and discussion as there is relevant background information in there.
3. The evaluation criteria is part of the methods… not results.
4. The first paragraph of section 3 has nothing to do with the paper and has been identified as present in another paper published by the authors. (ncRNA?). And in 4.2 RNA representation model? This makes no sense.
5. The results in 3.2 should be more descriptive in numbers. Just saying that one performs better or worse is not informative. Each part of the results section should address a section of the methods for a specific purpose.
6. There are results that are derived from methodology that has not been described. (E.g. Ablation experiments)
6. The datasets, markers and species evaluated could be shown as a Table. How many of each class for each species are available for each DNA methylation category? For example, if the classes are considerably imbalanced, PR-AUC should be used instead of ROC-AUC. Other metrics presented are threshold dependent.
7. It is not clear what software and packages were used for implementation. The authors should provide that along with the well-documented code and data used in the study. The authors can Github/Figshare or similar for that.
8.It is not clear how the training and cross-validation were done to then predict the test sets.
9. The authors describe the datasets and model architecture, but not exactly how the data was trained/validated. Can a model trained with one species predict well in another? Can subsets of data be used to train and predict a large test set? These are important questions that should be addressed.
10. There is no clear discussion, particularly taking into consideration the biological implications, limitations and/or future directions.
11. Overall, the authors should consider restructuring the writing of the MS to be more logical, precise and scientifically sound, and also easy to read to broad readship.
Author Response
First, we want to thank you for the valuable suggestions you provided for our paper. Your professional insights and detailed comments were crucial in helping us identify and improve the shortcomings of our work. Your feedback allowed us to present the research process and results more clearly, enhancing the overall quality and academic value of the paper. We are very grateful for your patience and thoroughness; your contributions have been of great help to our research. Thank you again for your support and guidance.

Reviewer 2 Report
Comments and Suggestions for Authors
In this work, the authors introduce DeepPGD, a deep learning framework to improve the identification of DNA methylation sites, a critical aspect of gene expression analysis, cellular differentiation, and disease progression studies. The framework utilizes a dual residual structure that integrates Temporal Convolutional Networks (TCN) and Bidirectional Long Short-Term Memory (BiLSTM) networks to capture complex DNA structural and temporal features. Although the performance of DeepPGD was evaluated through extensive experiments across various biological species, and the paper's topic fits well the journal's scope, many concepts need to be focused on improving the paper's quality:
- In the introduction section, I suggest highlighting your research contributions/novelty, trying to explain better the motivation for using deep learning approach; Moreover, Is the focus of the study on recognition/detection or classification? Please, clarify this aspect from abstract and introduction.
- Always in the introduction section, please improve the quality of the paper by making proper technical comparisons and highlighting better this research's limitations. Moreover, this section needs more details about similar studies by emphasizing at the same time the authors’ motivation for using deep learning in this scenario;
- Can the authors changing metrics performance approach situated in the results and discussions sections, to aligns with the methods described in the materials and methods section?
- Could the authors enhance the quality and clarity of Figure 10?
While the framework shows promising results across various species, further validation on more diverse datasets and under different experimental conditions is necessary to fully establish its generalizability and robustness.
- Despite the data used in this analysis being well described, the paper missing appropriate sources/links to the code to ensure the reproducibility of the results for other researchers. In addition, from a technical point of view, the part about using tools and libraries is missing in the methods section. This aspect is also necessary for the reproducibility of the experiments;
- In the experiments section, the manuscript lacks a comparison with and without data augmentation to ensure the model's applicability across diverse data. In addition, there may be concerns about how the model performs on complex datasets, although the text emphasizes generalizability. In addition, without details on how CX-RaysNet compares to existing state-of-the-art models, it's difficult to gauge its relative advantages or disadvantages;
- The experiments lack the DL studies, like the choice of hyperparameters of all DL models;
- Section "Conclusions" needs to be expanded to provide a more critical view of future detecting air leaks from fittings and the DL models associated with it, trying to update the research with the recent papers.
Comments on the Quality of English LanguageThe manuscript is generally well-written but would benefit from a thorough review to enhance clarity and readability. Improving sentence structure and coherence in certain sections, particularly the introduction and conclusion, will help in conveying the research contributions more effectively.
Author Response
First and foremost, we would like to extend our sincere thanks for the invaluable suggestions you provided for our paper. Your expert insights and thorough comments were essential in identifying and addressing the weaknesses in our work. Your feedback has greatly improved our ability to clearly present the research process and findings, thereby enhancing the overall quality and academic merit of the paper. We greatly appreciate your patience and meticulousness; your input has significantly benefited our research. Once again, thank you for your support and guidance.

Round 2
Reviewer 1 Report
Comments and Suggestions for Authors
The authors have adequately addressed the comments. However, regarding the code/data available on Github, I would recommend the authors to add comments/documentation. Please add a comprehensive README file.